# LOXL2 in Cancer: A Two-Decade Perspective

**DOI:** 10.3390/ijms241814405

**Published:** 2023-09-21

**Authors:** Amparo Cano, Pilar Eraso, María J. Mazón, Francisco Portillo

**Affiliations:** 1Departamento de Bioquímica UAM, Instituto de Investigaciones Biomédicas Alberto Sols, Consejo Superior de Investigaciones Científicas-Universidad Autónoma de Madrid (CSIC-UAM), 28029 Madrid, Spain; ampacano19@gmail.com (A.C.); peraso@iib.uam.es (P.E.); mazonmaria@gmail.com (M.J.M.); 2Instituto de Investigación Sanitaria del Hospital Universitario La Paz—IdiPAZ, 28029 Madrid, Spain; 3Centro de Investigación Biomédica en Red, Área de Cáncer (CIBERONC), Instituto de Salud Carlos III, 28029 Madrid, Spain

**Keywords:** LOXL2, human tumour sample, regulation, targets, mouse models, tumour progression

## Abstract

Lysyl Oxidase Like 2 (LOXL2) belongs to the lysyl oxidase (LOX) family, which comprises five lysine tyrosylquinone (LTQ)-dependent copper amine oxidases in humans. In 2003, LOXL2 was first identified as a promoter of tumour progression and, over the course of two decades, numerous studies have firmly established its involvement in multiple cancers. Extensive research with large cohorts of human tumour samples has demonstrated that dysregulated LOXL2 expression is strongly associated with poor prognosis in patients. Moreover, investigations have revealed the association of LOXL2 with various targets affecting diverse aspects of tumour progression. Additionally, the discovery of a complex network of signalling factors acting at the transcriptional, post-transcriptional, and post-translational levels has provided insights into the mechanisms underlying the aberrant expression of LOXL2 in tumours. Furthermore, the development of genetically modified mouse models with silenced or overexpressed LOXL2 has enabled in-depth exploration of its in vivo role in various cancer models. Given the significant role of LOXL2 in numerous cancers, extensive efforts are underway to identify specific inhibitors that could potentially improve patient prognosis. In this review, we aim to provide a comprehensive overview of two decades of research on the role of LOXL2 in cancer.

## 1. Introduction

Evolutionarily and structurally, the LOX family of enzymes in humans can be divided into two subfamilies. Subfamily 1 includes LOX and LOXL1, while subfamily 2 comprises LOXL2, LOXL3, and LOXL4 [1,2,3]. All members of the LOX family share a highly conserved carboxyl (C)-terminal amine oxidase catalytic domain, with identities ranging from 48% to 77% among them [4]. This domain contains a copper-binding motif and a lysine tyrosylquinone (LTQ) cofactor [5]. The Cu^2+^-binding site consists of three histidine residues (H626, H628, and H630 in LOXL2) and is crucial for LTQ biogenesis [3,6]. The LTQ cofactor is formed from conserved lysine and tyrosine residues via post-translational modification (K653 and Y689 in LOXL2). The LTQ cofactor is essential for the amine oxidase activity of LOXL2, which catalyses the oxidative deamination of the epsilon amino group of peptidyl lysine or hydroxylysine residues to produce highly reactive aldehydes, thereby establishing intra- or inter-cross linkages in collagen and elastin [6]. In addition, it has been reported that the LOXL2 amine oxidase catalytic domain can also deaminate unmethylated and trimethylated K4 in histone H3 and methylated K189 of TAF10 [7,8]. However, it should be noted that this deamination activity poses a conceptual challenge that needs further confirmation. The amino (N)-terminal region of LOX and LOXL1 contains a pro-sequence that is cleaved by BPM1 (LOX) or BMP1 and ADAMTS 14 (LOXL1) to generate the active mature enzymes [9,10]. In contrast, the N-terminal region of LOXL2-4 is characterised by the presence of four scavenger receptor cysteine-rich (SRCR) domains [5]. These SRCR domains may be involved in protein–protein interactions similar to SRCRs found in classical members of the scavenger receptor superfamily [11,12]. Previous studies have shown that LOXL2 SRCR domains interact with collagen IV and fibronectin [13] and that SRCR domain 1 is required for interaction with RNA binding proteins [14]. However, recent reports suggest new functions of the SRCR domains of lysyl oxidases. One study found that the SRCR domains of LOXL3 can deacetylate and deacetyliminate signal transducers and activators of transcription 3 (STAT3) on multiple acetyl-lysine sites [15]. Another study demonstrated that both LOXL2 and its LOXL2-Δe13 splice variant (lacking amine oxidase activity) directly catalyse the deacetylation of fructose-bisphosphate aldolase A (ALDOA) at K13 [16]. Finally, a third study revealed that LOXL2 associates with histone H3 and both the amine oxidase and the SRCR domains can catalyse H3K36ac deacetylation and deacetylimination [17].

Post-translational modifications are common among LOX enzymes. LOXL2, for instance, has three sites of N-glycosylation (N288, N455, and N644) essential for enzyme secretion [18,19], as well as seventeen disulphide bridges [20]. A low-resolution structure of the full-length LOXL2 obtained via X-ray scattering and electron microscopy [21], the crystal structure of a precursor form at a resolution of 2.4 Å [22] and a 3-D-predicted structure of the C-terminal amine oxidase domain of LOXL2 [23] are currently available. LOXL2 primarily exists as a monomer with some dimerization facilitated by the interaction between SRCR domains 1 and 2 [24].

LOXL2 mainly functions as an extracellular enzyme participating in the maturation and remodelling of the extracellular matrix (ECM) by catalysing the crosslinking of collagen and elastin fibres. Related to this function, its involvement in various physiological and pathological processes, such as fibrosis and cardiovascular diseases, has been well established [25,26,27,28,29]. However, in cancer, different factors and signalling pathways can lead to the abnormal expression and mislocalisation of LOXL2. The first report proposing LOXL2 as a pro-tumorigenic factor appeared in 2003 [30]. Since then, numerous studies have described LOXL2 overexpression in various tumour types significantly impacting patient prognosis [31] (refer to Section 2). Moreover, novel LOXL2 functions associated with cytoplasmic, perinuclear, and nuclear localisation have been discovered, with most of these functions being independent of its amine oxidase activity [13,15,16,17,32,33,34,35,36].

A very recent report has proposed a new LOXL2 function [37]. The study aimed at identifying differences in noncoding sequences between chimpanzees and humans. The authors found a deletion in a human LOXL2 promoter that alters the transcriptional output of LOXL2 via the loss of a SNAI2 binding site. The reintroduction of the conserved chimpanzee sequence into human cells provoked numerous myelination and synaptic function-related transcriptional changes. Based on these results, the authors propose LOXL2 as a gene controlling neuronal differentiation [37].

In this review, we highlight the significant progress made in the last two decades regarding LOXL2 expression patterns in tumour samples, the mechanisms regulating its expression levels, targets influencing tumour progression, and the valuable insights gained from genetically modified mouse models. Lastly, we discuss the potential prognostic and therapeutic values of LOXL2 in cancer.

## 2. LOXL2 in Human Tumour Samples

Over the last two decades, overexpression of LOXL2 has been consistently reported in numerous studies to be associated with tumour aggressiveness and poor prognosis in various types of cancer. Most of those studies have been covered in two recent reviews [1,38]. In recent years, a few reports have also conducted meta-analyses of available studies on correlations between LOXL2 overexpression and clinicopathological behaviour in tumours, primarily focusing on digestive system tumours and a few other types, such as oesophageal squamous cell carcinoma, breast cancer, and non-small-cell lung cancer [31]. These reports have further corroborated the link between LOXL2 overexpression and poor overall survival and worse clinicopathological characteristics of tumours. 

Additionally, over the last 4–5 years there has been an increase in the number of studies investigating LOXL2 expression in patient samples from a significant number of highly aggressive cancers of different origins: pancreatic adenocarcinoma [39,40,41], cervical carcinoma [42,43], osteosarcoma [44], oesophageal cancer [16], hepatocellular carcinoma [45], renal clear cell carcinoma [46], oral squamous cell carcinoma [47], and breast cancer [48,49,50] (summarised in Appendix A). In most cases, the overexpression of LOXL2 has been confirmed to be associated with poor prognosis and/or dissemination, reinforcing the role of LOXL2 as a prognostic biomarker in different tumour types.

Despite the wealth of information available on the overexpression of LOXL2 at the mRNA and/or protein level in tumours there are scarce data regarding the presence of genetic mutations of LOXL2 in human tumours that could potentially impact LOXL2 expression and/or function. To expand our understanding of LOXL2 in human tumours we utilised cBioPortal (www.cbioportal.org, accessed on 23 June 2023) to analyse the large TCGA PanCancer Atlas [51], which provides genetic information for 10,953 tumour samples to identify LOXL2 genetic alterations (Appendix A).

Regarding *LOXL2* gene copy alterations, gene amplification is observed in 0.11% of the samples (13 out 10,953) and deep deletions in 3% of the samples (317 out 10,953) (Appendix A). The frequency distribution of these genetic alterations among samples from different tumour types is shown in Figure 1.

Regarding to *LOXL2* point mutations, our screening revealed mutations in 1.37% of cancer samples (151 out 10,953), with 15 samples showing multiple LOXL2 mutations (Appendix A). Among the different tumours, skin cutaneous melanoma and uterine corpus endometrial carcinoma exhibited the highest frequency of LOXL2 mutations (8.60% and 6.43%, respectively) (Figure 2). This finding aligns with the high tumour burden typically associated with melanoma [52]. Regarding the localisation of the mutations, three of them were located in mRNA splicing sites and the remaining (148) were located within the ORF sequence. The LOXL2 ORF mutations fall into three categories: frame shift (3), nonsense (8), and missense (137) mutations (Appendix A). There were sixteen mutations that appeared in two different samples, seven in three samples and one in four of them (Appendix A). 

Most of the missense mutations do not affect key residues of LOXL2, that is N-glycosylation sites, copper binding sites, or LTQ precursor residues (Appendix A), except in the case of R338C, which eliminates a factor Xa cutting site (refer to Section 3.3). To investigate whether LOXL2 mutations could potentially impact its activity/function, we compared the localisation of LOXL2 missense mutations in the amino acid sequence alignment of LOXL2, LOXL3, and LOXL4 proteins. This analysis revealed that 36.5% (50 out 137) of the detected changes occur in fully conserved residues, leading us to speculate that they could potentially impact LOXL2 activity (Appendix A–G). However, experimental evidence will be necessary to validate this hypothesis due to the low amino acid alignment between the LOXL2, LOXL3, and LOXL4 proteins. 

In summary, the TCGA screening analysis suggests that LOXL2 mutational burden is not a major factor that globally affects the fitness of human tumours, although it is possible that specific mutations could be important in particular tumour types. In contrast, LOXL2 overexpression appears to be the main feature linked to tumour aggressiveness.

## 3. Control of LOXL2 Expression

Based on the inverse correlation found between elevated LOXL2 expression levels and overall survival, disease-free survival, and clinicopathological parameters in patients with different tumour types [1,31,38], numerous studies have focused on the regulation of LOXL2 expression in the last two decades. Most of those studies have been recently reviewed [38].

Figure 3 summarises the described factors operating on LOXL2 regulation at the transcriptional, post-transcriptional, and post-translational levels.

### 3.1. Transcriptional Regulation of LOXL2

Numerous studies highlight the role of hypoxia as a key transcriptional regulator of LOXL2 expression. Hypoxia controls LOXL2 expression in several ways. Hypoxia-inducible factor 1 (HIF1) binds to a hypoxia response element located in the *LOXL2* gene intron 1 and induces LOXL2 expression [53]. In addition, the dimer HIF1α/HIF1β recruits the lysine demethylase 4C (KDM4C) to the *LOXL2* promoter region. KDM4C demethylates the K9 of histone H3, thereby increasing the expression of LOXL2 [54]. Similarly, the upregulation of lysine demethylase 4B (KDM4B) by hypoxia also provokes the demethylation of trimethylated histone H3 at K9 at the *LOXL2* gene promoter, thus increasing LOXL2 expression [55]. Also, the upregulation of hypoxia-inducible factor 2α (HIF2α) by extracellular ATP through the P2Y2-AKT-PGK1 signalling pathway increases the expression of LOXL2 [56]. Moreover, hypoxia signalling directly controls ECM composition and remodelling [57], and ECM stiffness is another well-characterised factor controlling LOXL2 expression, which operates in two ways to upregulate *LOXL2* gene expression. In hepatocellular carcinoma cells, matrix stiffness upregulates LOXL2 through the integrin β1/α5-JNK-AP1 signalling pathway [58], while in M2 macrophages matrix stiffness activates the integrin β5-FAK-MEK1/2-ERK1/2 pathway, resulting in HIF1 upregulation and, consequently, an increase in LOXL2 expression [59]. The proto-oncogene c-FOS is another regulator of LOXL2 expression; c-FOS directly regulates the expression of the Wnt ligands Wnt7b and Wnt9a, which promote LOXL2 expression through the transcription factors ZEB1 and ZEB2 [44]. Recently, another study has described that Deubiquitinase zinc finger RANBP2-type containing 1 (ZRANB1) can deubiquitinate and stabilise SP1, causing an increase in LOXL2 expression [60].

*LOXL2* transcription is also regulated in different cancer scenarios by a series of factors whose mechanisms are still not fully characterised (summarised in Table 1).

### 3.2. Post-Transcriptional Regulation of LOXL2

An interrelated network of regulatory RNAs, involving microRNAs (miRNA), long noncoding RNAs (lncRNA), and circular RNAs (circRNA), converges to control LOXL2 expression at the post-transcriptional level. *LOXL2* mRNA levels are downregulated by miRNAs that bind to its 3′UTR region and, in some cases, the action of the miRNAs is counteracted by lncRNAs or circRNAs (Figure 3). 

Tumour-suppressive miR-26a/b, miR-29a/b/c, and miR-218 collectively downregulate *LOXL2* mRNA in head and neck squamous cell carcinoma and prostate cancer cell lines [76,77]. Additionally, the miR-26 family inhibits renal cancer cell migration and invasion by targeting *LOXL2* and procollagen lysine 2-oxoglutarate 5-dioxygenase 2 (PLOD2) mRNAs [78]. 

The miR-RNA-29 family members (miR-29a/b/c) also downregulate *LOXL2* mRNA in other tumour contexts [79,80,81,82]. However, in clear cell renal cell carcinoma, the lncRNA myocardial infarction-associated transcript (MIAT) binds to and inhibits miR-29c action on LOXL2 [83], while in osteosarcoma, the lncRNA human major histocompatibility complex p5 (HCP5) neutralises the action of miR-29b [84]. MiR-504 is another tumour-suppressive miRNA that targets *LOXL2* mRNA in non-small cell lung cancer cell lines [85]. 

In cervical cancer cells, circ0000228 sequesters miR-195 and upregulates LOXL2 expression to promote cervical cancer malignancy [86]. Similarly, lncKCNQ10T1 enhances *LOXL2* mRNA levels by blocking miR-1270, leading to decreased apoptosis of cervical cancer cells [87].

In colorectal cancer cells, LINC01347 enhances LOXL2 expression by competing with miR-328, resulting in an increase in cell proliferation and chemotherapy resistance [88]. On the other hand, prediction studies suggest that the lncRNA CARM-mirR-192-LOXL2 axis is associated with poor overall survival, immune infiltration, and immune checkpoint expression in hepatocellular carcinoma [89]. 

Although not related to the cancer field, it is worth mentioning that in a cell model of hypertrophic flavum ligamentum, *LOXL2* mRNA is targeted by miR-4731, and LOXL2 downregulation is relieved by circPDK1 [90].

A recent report also suggests that selective splicing of *LOXL2* mRNA could also be a new source of post-transcriptional regulation [38].

### 3.3. Post-Translational Regulation of LOXL2

Figure 3 illustrates the mechanisms described so far that act post-translationally on LOXL2. 

Extracellular LOXL2 undergoes two proteolytic cleavages catalysed by paired basic amino acid cleaving enzyme 4 (PACE4) and factor Xa. PACE4 cleaves LOXL2 between the SRCR2 and SRCR3 domains (site K317↓). The importance of this proteolytic processing on the amine oxidase activity of LOXL2 is controversial. One report suggests that the proteolytic processing is not essential for the amine oxidase activity in solution or crosslinking of collagen type IV in ECM [91], while another report suggests that it does not affect LOXL2-mediated crosslinking of soluble collagen type IV in vitro, but is essential to crosslink insoluble collagen IV within the ECM [92]. Factor Xa cleaves LOXL2 at the beginning of SRCR3 (site R338↓). LOXL2 processing by factor Xa results in reduced cross-linking activity in the ECM and a change in LOXL2 substrate preference from collagen type IV to type I collagen [93]. 

LOXL2 interacts with extracellular heat shock protein HSP90 in breast cancer and glioma cell lines, although the functional significance of this interaction remains to be elucidated [94]. 

In endothelial cells, LOXL2 is negatively regulated at two levels. On the one hand, secreted epidermal growth factor-like protein 7 (EGFL7), a regulator of vascular elastogenesis, binds to the LOXL2 catalytic domain, preventing the conversion of tropoelastin into mature elastin [95]. On the other hand, the hypoxia-induced long noncoding antisense transcript of GATA6 (lncRNA GATA6-AS) interacts with nuclear LOXL2, interfering with its oxidative deaminase activity and inhibiting the removal of activating H3K4me3 chromatin marks, thereby regulating endothelial gene expression [96]. 

Another negative regulator of LOXL2 is the tripartite motif-containing protein 44 (TRIM44), a regulator of tumour immunity in gastric cancer. TRIM44 interacts with LOXL2 in the cytoplasm and regulates its stability through the ubiquitin–proteasome pathway. The reduction in the amount of LOXL2 interferes with ECM remodelling and influences the immunity of the tumour [97].

In ovarian granulosa cells, LOXL2 is phosphorylated by the large tumour suppressor kinase 1 (LATS1), a negative regulator of YAP in the Hippo signalling pathway, although the consequences of this phosphorylation in the context of cancer have not yet been analysed [98].

All these findings provide insights into the intricate regulatory network governing LOXL2 expression and activity in various cancer types. Understanding the mechanisms controlling LOXL2 expression and function can potentially lead to the development of targeted therapies for cancer treatment in the near future.

## 4. Targets of LOXL2 in Cancer

Nowadays, numerous studies have firmly established the implication of LOXL2 in the progression and metastasis of various tumour types. LOXL2 interacts with extracellular, cytoplasmic, and nuclear targets, and some of its actions are independent of its amine oxidase catalytic activity [13,15,16,17,32,33,34,35,36]. The mechanisms involved in the cytoplasmic and nuclear localisation of LOXL2 are still unresolved questions that warrant future investigation.

### 4.1. Extracellular Targets

Extracellular LOXL2 exerts its effects on tumour cells and the tumour microenvironment at different levels (Figure 4).

In breast tumour cells, secreted LOXL2 promotes ECM remodelling leading to increased stromal stiffness. This, in turn, activates cancer-associated fibroblasts through integrin-mediated focal adhesion kinase (FAK) signalling [99,100,101]. In oral tumour cells, secreted LOXL2 oxidises lysine residues in platelet-derived growth factor receptor beta (PDGFRβ) on stromal fibroblasts, enhancing platelet-derived growth factor (PDGF-AB) signalling and promoting stromal fibroblast proliferation via ERK activation [102]. In both cases, tumour-secreted LOXL2 activates surrounding fibroblasts to create a supportive local niche. In breast cancer cells, secreted LOXL2 enhances lymphatic endothelial cell invasion through AKT and ERK signalling and stimulates fibroblasts to secrete pro-lymphangiogenic factors such as vascular endothelial growth factor C (VEGF-C) and stromal cell-derived factor 1 (SDF-1α) [50].

Tumour-secreted LOXL2 can also contribute to the early steps of metastatic colonisation at distant organs. Once LOXL2 reaches distant organs, either as a soluble enzyme or packaged in exosomes, it can remodel the ECM and stimulate premetastatic niche formation. This involves recruiting bone marrow progenitor cells and the transcriptional regulation of fibronectin and various cytokine expressions [103,104,105,106,107].

### 4.2. Cytoplasmic Targets

Intracellular LOXL2 exerts its pro-tumorigenic role by modifying different cytoplasmic targets and regulating various signalling pathways in different cancer systems (Figure 5).

In normal mammary epithelial cells, LOXL2 overexpression induces oncogenic transformation and cancer progression by activating the Erb-B2 receptor tyrosine kinase 2 (ERBB2) through the production of reactive oxygen species (ROS) [108].

Intracellular LOXL2 also influences cytoskeleton dynamics and cell motility capabilities to promote tumour migration and metastasis. It does so by activating the FAK signalling pathway [35,109,110], increasing the phosphorylation and activation of ezrin [33], and interacting with vimentin [32]. Additionally, LOXL2 can promote tumour metastasis by regulating the levels of phosphorylated AKT through mechanisms that are not yet fully understood [111] and by stabilising integrin subunits α5 (ITGA5) and β1 (ITGB1) [112].

LOXL2 also affects endoplasmic reticulum (ER) homeostasis. Its overexpression indirectly activates the unfolded protein response (UPR), leading to epithelial-to-mesenchymal transition (EMT). EMT is a genetic and reversible program that leads to the loss of epithelial status, as well as to the gain of mesenchymal traits, resulting in cells with a greater capacity for mobility, migration, and invasion [113]. Several transcription factors (TFs) have been described as EMT inducers (EMT-TFs), including SNAI1, SNAI2, ZEB1, ZEB2, TCF3 (also known as E47), and TWIST1 [114,115,116]. Overexpressed LOXL2 is retained in the ER, where it interacts with heat shock protein family A (Hsp70) member 5 (HSPA5), or BiP, a regulator of the UPR. This interaction activates the IRE1-XBP1 signalling pathway of the UPR, leading to the expression of EMT-TFs such as SNAI1, SNAI2, ZEB2, and TCF3, promoting EMT [34].

LOXL2 can also modulate other cancer hallmarks. It influences tumour chemosensitivity by enhancing autophagy. LOXL2 increases *ATG7* expression possibly by promoting the phosphorylation of ERK1/2 through an unknown mechanism [117]. LOXL2 interactions with the N-terminal domain of myristoylated alanine-rich C kinase substrate-like 1 (MARCKSL1) inhibits MARCKSL1-induced apoptosis [118]. Moreover, LOXL2 affects immune cell infiltration and immune checkpoints through its interaction with IQ motif-containing GTPase-activating protein 1 (IQGAP1) [45]. In addition, LOXL2 catalyses the deacetylation of fructose-bisphosphate aldolase A (ALDOA) at K13, leading to the mobilisation of aldolase A from the cytoskeletal to the cytosolic fraction, which enhances glycolysis and subsequently promotes tumour progression [16].

### 4.3. Nuclear Targets

Nuclear LOXL2 is primarily an EMT-inducing factor that acts at the transcriptional and post-transcriptional levels on various transcription factors and histone H3 marks impacting EMT (Figure 6).

Nuclear LOXL2 can induce EMT thorough different mechanisms. Both LOXL2 and catalytically inactive mutants can interact with SNAI1 to repress *E-cadherin* gene (CDH1) expression by blocking SNAI1 GSK3β-dependent degradation [119,120]. LOXL2 is recruited by SNAI1 to heterochromatin, where it oxidises the trimethylated K4 in histone 3 (H3K4me3), thereby repressing *CDH1* transcription [7,121]. However, it is worth noting that this deamination activity poses a conceptual challenge that needs further confirmation. LOXL2-dependent H3K4me3 oxidation also leads to chromatin compaction, reduced DNA damage response, and increased resistance to anticancer drugs [48]. The repression of E-cadherin expression is also mediated by the interaction between LOXL2 and TCF3/E47 and their direct binding to the *CDH1* promoter region [107]. Additionally, LOXL2/E47 collaborate in the direct transcriptional regulation of fibronectin (FN) and the cytokines tumour necrosis factor (TNFα), angiopoietin 1 (ANGPT1), and colony-stimulating factor 2 (CSF2), which contribute to premetastatic niche formation [107]. LOXL2 also represses the transcriptional expression of components of tight junctions and cell polarity complexes, including claudin-1 (CLDN1) and LLGL scribble cell polarity complex component 2 (LLGL2) genes [110]. Since LOXL2 cannot bind directly to DNA [35], this regulation must be mediated by an as-yet-unidentified transcription factor. In skin cancer cells, LOXL2 is recruited by KLF transcription factor 4 (KLF4) to the notch receptor 1 (NOTCH1) promoter, where it decreases H3K4me3 levels, thereby impairing RNA polymerase II recruitment and inhibiting *NOTCH1* transcription, thus repressing epidermal differentiation [122]. The nuclear action of LOXL2 in the invasion and tumorigenesis of breast cancer cells is also mediated, at least in part, through the upregulation of the receptor activity-modifying protein 3 (RAMP3) gene, although the implication of RAMP3 in EMT has not been investigated yet [123]. 

In contrast to the above-described pro-tumorigenic action of LOXL2, in uterine endometrial carcinoma LOXL2 appears to play the opposite role. LOXL2 associates with and catalyses H3K36ac deacetylation thus blocking H3K36ac-dependent transcription of genes, including MYC proto-oncogene, BHLH transcription factor (MYC), cyclin D1 (CCND1), HIF1A, and CD44, thereby restricting cancer cell proliferation [17].

Recently, several reports have implicated nuclear LOXL2 in functions beyond EMT. LOXL2 positively affects HIF1, thus establishing a positive feedback loop. In pancreatic ductal adenocarcinoma LOXL2 stabilises HIF1 by inhibiting HIF1 hydroxylation and enhancing the expression of HIF1, increasing the transcription of multiple glycolytic genes, thereby promoting aerobic glycolysis (Warburg effect) and pancreatic adenocarcinoma progression [40]. In hepatocellular carcinoma, however, LOXL2 increases *HIF1* gene expression by counteracting, in an SNAI1-dependent way, the negative action of fructose-1,6-bisphosphatase (FBP1) on *HIF1* gene expression, resulting in increased aerobic glycolysis and angiogenesis [124]. Moreover, it has recently been observed that blockade of the HIF1-LOXL2 signalling pathway alleviated tumour immunosuppression [125]. In addition to EMT, nuclear LOXL2 also promotes angiogenesis through interaction with GATA binding protein 6 (GATA6) and upregulation of vascular endothelial growth factor A (VEGFA) gene expression [126]. On the other hand, high levels of LOXL2 inhibit the 5-FU-induced apoptosis of colorectal cancer cells. In this case, LOXL2 activated the Hedgehog signalling pathway by promoting the expression of smoothened, frizzled class receptor (SMO) and GLI family zinc finger 1 and 2 (GLI1, GLI2) factors [127]. Also, LOXL2 promotes gene expression of the anti-apoptotic proteins baculoviral inhibitor of apoptosis protein (IAP) repeat-containing 3 (BIRC3) and murine double minute 2 (MDM2) in hepatocellular carcinoma [128].

Furthermore, it has recently been described that nuclear LOXL2 interacts with RNA binding proteins involved in all aspects of mRNA metabolism, many of them impinging on EMT. Although this finding requires further study, it may lead to the expansion of the pathways that LOXL2 uses to modulate tumour progression [14].

In summary, LOXL2 involvement in various types of cancer is multifaceted, as it targets different cellular compartments and participates in numerous signalling pathways. Understanding these intricate regulatory mechanisms can offer valuable insights for the development of targeted therapies for cancer treatment.

## 5. Genetically Modified Mouse Models

The use of genetically engineered mouse models (GEMMs) has been instrumental in understanding the role of specific gene products in various pathologies and advancing our knowledge of their underlying molecular mechanisms. In the case of lysyl oxidase proteins, GEMMs for all five family members have been generated over the last two decades (summarised in Table 2).

Most of these models have been generated as germ line/constitutive GEMM in which the gene of interest has been either deleted (knockout, KO) or overexpressed (knockin, KI) in the whole embryo or all adult tissues. Depleting individual Lox/Loxl genes in constitutive or conditional adult KO models leads to diverse phenotypes ranging from perinatal lethality to alterations in ECM remodelling, or varied changes in several tissues and organs affecting the development of the cardiovascular, muscle–skeletal, or lung systems; or causing hepatic distension, among other changes. The specific phenotypic outcomes depend on the particular Lox/Loxl gene that has been modified [17,122,129,130,132,133,135,136,137,138,139,140,141]. A few constitutive KI GEMMs have also been generated for Lox and Lolx2 genes, as well as for the Loxl2-Δe13 (L2ΔE13) splice variant, presenting milder phenotypes [16,122,131,134] (Table 2).

Regarding GEMMs related to cancer, valuable information has been obtained from Loxl2 and Loxl3 models. For Loxl2 our group has generated constitutive and conditional Loxl2-KO (L2-KO) and Loxl2-KI (L2-KI) models in recent years [39,122,142]. Other groups have recently generated one additional constitutive L2-KO [17] and one L2Δe13-KI splice variant model [16] (Table 3). As for Loxl3, our group has recently described one conditional Loxl3-KO (L3-KO) [143]. In most of the constitutive models, no significant phenotypic alterations in the adult phenotype were detected, apart from perinatal lethality found in about 60% of homozygous L2-KO mice, and male sterility in 90% of homozygous L2-KI mice [122], and alterations in the female reproductive system in another L2-KO model [17] (Table 2).

The first cancer study describing the use of constitutive L2-KO and L2-KI GEMMs was reported in 2015 [122]. In this study, L2-KO and L2-KI mice, along with corresponding controls, were exposed to the experimental system of two-step mouse skin carcinogenesis. This process involved the topical application of one simple dose of the carcinogen DMBA (7,12-dimethylbenz(a)anthracene) followed by sequential doses of TPA (12-O-tetradecanoylphorbol-13-acetate) for up to 30 weeks, resulting in the development of premalignant skin lesions, some of which could progress to malignant squamous cell carcinomas [144]. The use of both complementary L2-KO and L2-KI GEMMs allowed us to demonstrate a key role for Loxl2 in tumour initiation and progression in mouse and human squamous cell carcinomas. When exposed to the DMBA/TPA treatment, the skin of L2-KI mice significantly increased in tumour burden and malignant progression, with reduced tumour latency compared with controls. In contrast, the same treatment in the skin of L2-KO mice resulted in the opposite phenotype, i.e., decreased tumour lesion size and reduced rate of malignant progression compared with controls.

Further mechanistic studies in mouse and human cell lines revealed that LOXL2 negatively regulates epidermal differentiation and the Notch1 signalling pathway in premalignant skin lesions. The study reported for the first time that Loxl2 is a transcriptional repressor of *NOTCH1* gene expression, acting in coordination with the KLF4 transcription factor [122]. This finding was particularly relevant in human head and neck squamous cell carcinomas and cervical squamous cell carcinomas where NOTCH1 acts a tumour suppressor [122]. The study further confirmed our previous findings, which linked intracellular LOXL2 overexpression as a prognostic marker of larynx squamous cell carcinoma [145]. 

The recently described constitutive L2ΔE13-KI splice variant model has been crucial in identifying the novel deacetylase activity of Loxl2 and the L2ΔE13 variant, which promotes metabolic reprogramming and tumour progression in oesophageal cancer [16]. Notably, this study also reported that overexpression of the LOXL2/L2ΔE13 variant and decreased acetylation of aldolase A at K13 residue (one of the deacetylating targets of LOXL2) served as predictors of poor clinical behaviour in oesophageal cancer patients [16]. 

A recent study described the generation of another constitutive L2-KO GEMM, using the same Loxl2-KO allele strategy used in the previous L2-KO model [122], but in a different genetic background [17]. In this case, in contrast to the previous reports, female L2-KO mice spontaneously developed uterine hypertrophy and uterine carcinoma, which was associated with novel deacetylation and deacetylimination activity of Loxl2 on histone H3 at residue K36 (H3K36ac) [17]. Interestingly, low LOXL2 and high H3K36ac levels were associated with poor prognosis in uterine endometrial patients, suggesting nuclear LOXL2 as a protective factor against uterine cancer development [17].

Further studies have described the generation of conditional L2-KO and L2-KI GEMMs in two different cancer types: breast cancer and pancreatic adenocarcinoma. In breast cancer, two complementary L2-KO and L2-KI transgenic mouse models were generated in PyMT-induced breast cancer in the mammary gland by crossing L2-KO or L2-KI alleles with the MMTV-Cre; PyMT^+/−^ mice. These models provided functional evidence that LOXL2 is a key driver of breast cancer metastasis [142]. Loxl2 deletion in mammary tumour cells dramatically decreased lung metastasis, while Loxl2 overexpression promoted metastatic tumour growth of MMTV-PyMT-breast tumours. Mechanistic studies in primary cell lines derived from PyMT-breast tumours identified the association of Loxl2 with increased levels of Snai1 and several cytokines that promote the generation of a premetastatic niche [142]. This study also confirmed previous findings that implicated Loxl2 in the generation of the premetastatic niche using syngeneic breast cancer cell models [107] and the association of perinuclear LOXL2 overexpression with lung metastasis in human basal-like breast cancer, a subtype of the highly aggressive triple negative breast tumours [110]. In the case of pancreatic adenocarcinoma, four conditional L2-KO and L2-KI GEMMs were generated by crossing conditional L2-KO and L2-KI alleles with two broadly studied pancreatic adenocarcinoma mice models: KPC (K-Ras^+/LSL-G12D^; Trp53^LSL-R172H^; Pdx1-Cre) or KC (K-Ras^+/LSL-G12D^; Pdx1-Cre) mice lines. This resulted in the generation of KPC-L2KO or KC-L2KO, and KPC-L2KI or KC-L2KI mice [39]. Detailed analysis of the clinicopathological parameters and cellular and molecular characterisation of the four models and derived cell lines established that Loxl2 depletion significantly decreased metastasis and increased overall survival without affecting primary tumour development and growth. In contrast, Loxl2 overexpression promoted primary and metastatic tumour growth and decreased overall survival [39]. These results agree with the association found between LOXL2 overexpression and poor prognosis in human pancreatic adenocarcinoma described in that study and other reports [39,40,41]. Interestingly, Alonso-Nocelo et al. identified that both extracellular functions (ECM remodelling and collagen crosslinking) and intracellular actions of Loxl2 (EMT induction and increased stemness of tumour cells) contribute to the metastatic effect of Loxl2 ablation and Loxl2 overexpression in the pancreatic adenocarcinoma models, respectively [39], highlighting the complex in vivo actions of Loxl2 in pancreatic tumour progression. Moreover, the study identified oncostatin M secreted by tumour-associated macrophages as an inducer of LOXL2 expression [39].

Regarding Loxl3, the conditional L3-KO model generated in the context of mouse melanoma (Table 3) has been instrumental in supporting a key action of Loxl3 in melanomagenesis and lymphatic dissemination of cutaneous melanoma [143], which agrees with the high expression levels of LOXL3 found in association with SNAI1 and other EMT factors in human melanoma samples and cell lines [143,146]. 

Apart from GEMMs, numerous gain and loss of function studies of LOXL2 in different human cancer cell lines tested in xenografted models in immune compromised mice have been reported in the last two decades. Some syngeneic cancer cell models have also been reported in which the expression of Loxl2 has been genetically manipulated and the resulting cell lines orthotopically injected into immune-competent syngeneic mouse lines, providing valuable preclinical models. For example, breast cancer [107] and oral squamous cell carcinoma [102] syngeneic models have provided mechanistic insights into the molecular action of Loxl2 in tumour progression and metastasis. Moreover, some of those syngeneic or specific GEMMs have been used to test new LOXL2 inhibitors or specific LOXL2 antibodies in in vivo contexts and to analyse the influence of LOXL2 in the response to anti-immune therapies [17,44,102,147].

The various Loxl2-GEMMs described above can provide highly valuable preclinical models for the in vivo testing of novel LOXL2 inhibitors or specific LOXL2 antibodies in the near future. 

## 6. Prognostic and Therapeutic Value of LOXL2 in Cancer

As evidence for the high expression of LOXL2 in several types of cancer has increased, the possibility of considering LOXL2 as a potential prognostic marker has also grown. Numerous examples of the presence of LOXL2 in serum samples of fibrosis patients have pointed to LOXL2 as a potential biomarker for idiopathic pulmonary fibrosis [148], cardiac fibrosis [149], and other pathologies [150,151].

Regarding the use of LOXL2 as a prognostic marker in cancer, few attempts have been made to date. The strategy directed to evaluate changes in the proteome of extracellular vesicles seems promising. In this sense, the evaluation of LOXL2 levels in serum exosome fractions from head and neck squamous cell carcinoma patients supported the correlation between elevated LOXL2 and low-grade tumours [152].

During the maturation of LOXL2, the release of the signal peptide generates a LOXL2 neo-epitope that can be measured with ELISA [153]. This novel and specific ELISA assay appears to be another approach for detecting elevated LOXL2 levels in fibrosis or cancer. However, it is worth noting that the high concentration of the neo-epitope required for detection in the ELISA assay raises a cautionary note about its current utility. In summary, early cancer identification based on the use of LOXL2 as a biomarker may be of potential benefit in designing treatments and improving clinical outcomes, but further investigation and additional trials are clearly needed.

Given the demonstrated role of LOXL2 in tumour progression and fibrotic disease, inhibiting the classical catalytic activity of the enzyme appeared as a potential therapeutic alternative. The recent discoveries of different activities of LOXL2 and LOXL2 actions independent of its oxidative deamination catalytic activity in cancer have opened the way for novel therapeutic approaches. Several strategies have been followed to inhibit LOXL2 activity or action (reviewed in [154,155,156] (Figure 7)).

### 6.1. Antibodies

As an approach to inhibit LOXL2, antibodies against the protein appeared to be the first option. The anti-LOXL2 murine monoclonal antibody AB0023 was one of the first antibodies shown to inhibit LOXL2 enzymatic activity in vitro. This antibody binds to the fourth SRCR domain of the protein, making it an allosteric inhibitor [156]. This is an interesting feature, as additional enzymatic functions have been associated with the SRCR domains of the lysyl oxidases [13,15,16,17].

AB0023 has been tested in mice, in combination with taxol, and shown to inhibit tumour angiogenesis in highly angiogenic tumours derived from ovarian carcinoma and lung carcinoma cells. Although AB0023 did not inhibit tumour development, it promoted enhanced sensitivity to treatment with taxol, suggesting a possible beneficial use in angiogenic diseases by improving the delivery of chemotherapy [157]. Moreover, AB0023 was effective in vivo in a murine model of pancreatic cancer, reducing tumour collagen density [41], and also in xenograft models of both primary and metastatic cancers and in fibrosis models of the liver and lung [158].

Following these promising results with AB0023 in animal models, a humanised version of the antibody, named Simtuzumab was developed [159]. This new antibody was used in several randomised phase II clinical trials for different pathologies but in none of them the antibody improved the clinical outcomes of the patients [160,161,162,163,164,165]. More recently, Findlay et al. [166] demonstrated the effective inhibition of LOXL2 enzyme activity in vitro, in vivo, and in a phase I trial in healthy humans using a novel small molecule, PXS-5338. This was achieved using an activity-based probe known as PXS 5878. Under the same conditions, AB0023 was shown to be a partial and low potency inhibitor of recombinant human LOXL2. These new findings provide a plausible explanation for the clinical failure of Simtuzumab. GS341 is another anti-LOXL2 antibody developed in mice [167]. This antibody targets the active site of LOXL2 and was shown to alter the orientation and thickness of fibrillary ECM proteins due to the inhibition of LOXL2. The disrupted collagen morphologies caused by GS341 interfered with the adhesion and invasion properties of human breast cancer cells in in vitro assays [167]. However, no humanised version of GS341 has been reported yet and it is not clear if it will show the same improvement in patients as observed in animal models.

Targeting aberrant signalling in the tumour microenvironment with specific antibodies could be a focus in many treatments. One caveat to this strategy is the growing evidence implicating intracellular LOXL2 in a wide variety of processes.

### 6.2. Copper Chelators

Since lysyl oxidases need copper to be active, the use of copper chelators has been considered an effective strategy to inhibit lysyl oxidases and block cancer development. Some initial studies pointed to an inhibition of tumour angiogenesis by reducing copper availability [168]. The use of tetrathiomolybdate or D-penicillamine showed promising results [169,170,171,172]. Nevertheless, the fact that copper ions are implicated in many enzymatic reactions and biological processes has relegated the use of copper chelators and they are presently being replaced by new inhibitors.

A different approach to inhibit LOX activity is based on the design of two peptides, called M peptides, corresponding to the copper-binding region in the LOX protein [173]. The two peptides inhibited hypoxia-induced extracellular LOX activity and inhibited hypoxia-induced endothelial tube formation and the migration of HuVEC cells. A similar approach could be useful for other members of the LOX family. However, in vitro and in vivo studies of the effect of this strategy on cancer have not been performed yet.

### 6.3. Small Molecule Inhibitors of LOXL2

The structure, specificity, and IC50 of small-molecule inhibitors of LOXL2 are summarised in Appendix A.

BAPN (β-Aminopropionitrile) is the first described inhibitor of lysyl oxidases [174]. BAPN was found to bind the LOX protein through a covalent bond [175]. The structure of BAPN contains a nitrile group and a primary amine which mimics the role of the peptidyl lysine that reacts with the LTQ factor during the oxidative deamination reaction. Through this amine group, BAPN forms a stable product with the LOX protein, thus preventing the recycling of the LTQ cofactor and irreversibly inhibiting the enzyme [176]. Difficulties in improving the chemical structure of BAPN, complicating its preclinical optimisation, opened the way to search for other small-molecule inhibitors.

Several compounds have been developed in recent years with LOX and LOXL2 inhibitory activity. One of them, the haloallylamine-based molecule PXS-S1A, is an example of this new generation of inhibitors. PXS-S1A displays an almost identical inhibitory capacity and selectivity against LOXL2 and LOX. A later chemical modification led to PXS-S2A, which is highly selective for LOXL2. Both inhibitors inhibited cell proliferation in 2-D and 3-D assays and were effective in 3-D spheroid-based invasion assays. A difference between the two inhibitors was seen in migration assays, where PXS-S2A showed a reduced effect in wound closure as compared with PXS-S1A, suggesting that both LOXL2 and LOX are important in 2-D migration. Metastatic dissemination was also evaluated for the two PXS inhibitors and it was found that LOXL2 enzymatic inhibition was not sufficient to inhibit metastasis, indicating that there may be important non-enzymatic roles for LOXL2 in metastatic dissemination [177]. PXS-5153A is a fluoroallylamine-bearing irreversible and dual inhibitor for LOXL2 and LOXL3 that has been used to elucidate the role of these two enzymes in models of collagen crosslinking and fibrosis [178]. 

A potent and highly selective LOXL2 inhibitor, PAT-1251, based on a benzylamine that is 2-substituted with pyridine-4-ylmethanamines, was described [179], and shown to be more than 400-fold selective over LOX. 

A new pharmacological inhibitor of LOX, CCT365623, an amino methylene thiophene (AMT)-based compound, is more potent than BAPN and inhibits both LOX and LOXL2 [147]. Structure−activity relationships of AMT inhibitors led to a series of derivatives, with improved potency towards LOXL2 inhibition versus LOX. One of them, named 21b, was assayed in a PyMT mouse model of breast cancer and promoted a delay in primary tumour development and a reduction in tumour growth rate [147]. 

A common feature of the small molecules developed as LOXL2 inhibitors is the presence of a primary amino group. This group competes with the epsilon amino group of lysine residues in LOXL2 substrates during their interaction with the LTQ cofactor, enabling specific binding to the enzyme.

In a different approach, Wei et al. performed phenotypic screens for small molecules that could inhibit TGF-β1 signalling, a driver of collagen accumulation and fibrotic disease. The screening led to the identification of corilagin, a trihydroxyphenolic compound, which induces the auto-oxidation of K731 and irreversibly inhibits LOXL2. A product of this reaction is an inhibitor of the TβR1 kinase. The inhibition of LOXL2 and the receptor kinase resulted in the blockade of pathological collagen accumulation in vivo, opening the way for a therapeutic approach to attenuate fibrosis [180].

Very recently, an interesting approach to search for new inhibitors of LOXL2 has been used [181]. The strategy mainly relies on two ideas, the first being the study of already FDA-approved drugs to repurpose their applications. The second idea is to use the crystal structure of hLOXL2 to identify potential druggable cavities or pockets in which previously selected drugs, based on 2-D and 3-D structures as well as molecular weight, could “fit”. This approach uncovered levoleucovorin as an efficient and stable LOXL2 inhibitor that exhibits antiproliferative efficacy in two breast cancer cell lines [181]. Nevertheless, further preclinical investigation of these inhibitors is clearly needed.

### 6.4. Inhibitors of LOXL2 Transcription

Several plant-derived compounds have demonstrated their ability to negatively regulate LOXL2 expression and could represent a new therapeutic approach. 

Escin Ia is obtained from the fruits of *Aesculus chinensis Bunge*. Escin Ia inhibits the invasion and migration of MDA-MB-231 breast cancer cells through selective downregulation of *LOXL2* expression and prevents the EMT process. When tested in xenograft mice models Escin Ia suppressed MDA-MB-231 metastasis [182]. 

Salidroside is isolated from Rhodiola rosea and reduces *HIF-1α* and *LOXL2* expression levels in the human pancreatic cancer cell line BxPC-3, inhibiting its proliferation and invasion under hypoxia. In xenograft assays of BxPC-3 cells, treatment with salidroside reduces tumour volume, increases apoptosis, and reduces metastatic infiltration in the lungs [183].

Dihydroartemisinin (DHA), a derivative of the Chinese medicine artemisinin, was shown to enhance antiangiogenic drug-induced toxicity in osteosarcoma cells. DHA significantly inhibited osteosarcoma cell proliferation, migration, and invasion and induced apoptosis through the downregulation of *LOXL2* expression [184].

The detailed molecular mechanism used by these compounds to inhibit the expression of LOXL2 and their degree of specificity is something that remains to be studied.

In conclusion, LOXL2 is a promising target for cancer therapy and the development of anti-LOXL2 drugs appears feasible. Although preclinical studies with various small molecule inhibitors are encouraging, more clinical trials are required to confirm their effectiveness.

## 7. Future Perspectives

Two decades after the first mention of the pro-tumorigenic role of LOXL2, it has been firmly established that increased expression of LOXL2 in many types of cancer significantly impacts patient prognosis. Currently, it is known that multiple pathways can result in altered LOXL2 expression, and that LOXL2’s effects on various targets within different cellular compartments can enhance cancer capabilities. Furthermore, the modulation of LOXL2 targets can occur through mechanisms independent of its amine oxidase activity. The diverse cellular localisations of LOXL2, coupled with its various mechanisms of action pose significant challenges in developing therapeutic agents aimed at targeting LOXL2 to block tumour progression. It may be necessary to utilise distinct therapeutic drugs designed to inhibit the SRCR domains in addition to the C-terminal oxidase domain of the enzyme. Alternatively, an approach to control the LOXL2 expression level in tumours could involve the development of drugs specifically targeting LOXL2 expression. Another future challenge lies in unravelling the mechanisms by which LOXL2 reaches intracellular compartments, particularly the nucleus, as this could present new therapeutic opportunities. In addition, the potential functional redundancy among different LOX proteins may require the inhibition of additional family members alongside LOXL2 to enhance the therapeutic strategies related to this family of proteins.

## Figures and Tables

**Figure 1 ijms-24-14405-f001:**
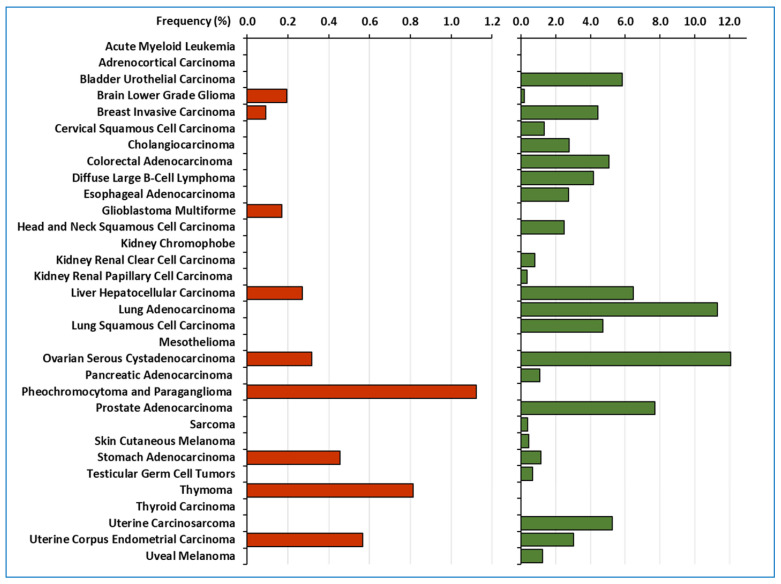
Frequency of gene copy alterations detected in *LOXL2* locus among all tumour types. Red bars (**left**) correspond to the frequency of gene amplification and green bars (**right**) to deep deletions.

**Figure 2 ijms-24-14405-f002:**
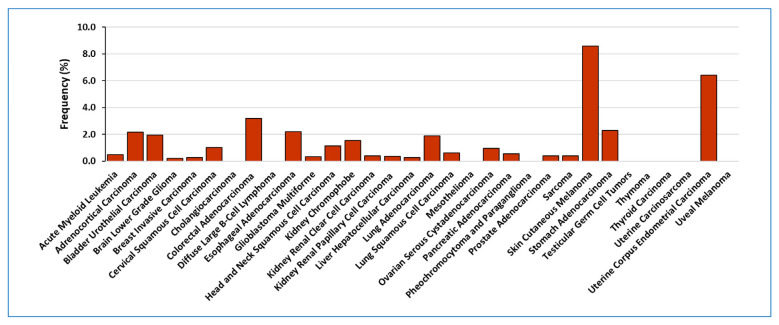
Frequency of point mutations detected in *LOXL2* gene among all tumour types.

**Figure 3 ijms-24-14405-f003:**
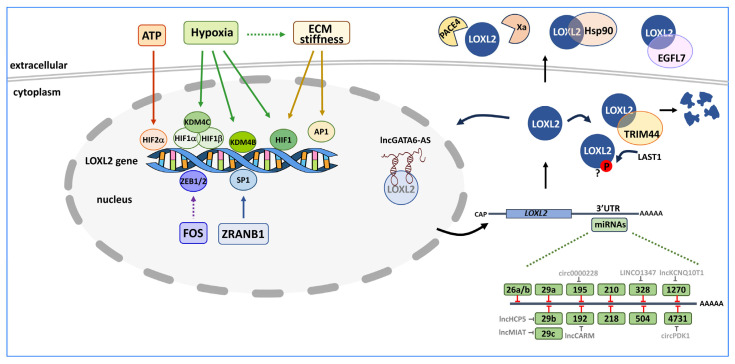
LOXL2 regulation. Left side, *LOXL2* gene expression is regulated in different cancer scenarios by three well characterised signalling pathways (extracellular ATP, hypoxia, and ECM remodelling) that impinge on different transcriptional factors, including HIF1α, HIF1β, HIF2α, and lysine demethylases KDM4B and KDM4C. The proto-oncogene c-FOS controls *LOXL2* expression through the Wnt7/9-ZEB1/2 axis. Deubiquitinase ZRANB1 stabilises the transcription factor SP1. Right bottom cytoplasm side, several miRNAs act through the *LOXL2* 3’UTR mRNA region to downregulate its gene expression. Long noncoding RNAs (lncRNA) and circular RNAs (circRNA) counteracting the miRNAs (green boxes) are marked in grey. Right upper cytoplasm side, LOXL2 is directed to the ubiquitin–proteasome pathway by the interaction with TRIM44. LOXL2 is phosphorylated by LAST1 with unknown functional consequences. Upper right side, LOXL2 in the extracellular compartment undergoes proteolytic processing by PACE4 and factor Xa proteases, and secreted EGFL7 inhibits LOXL2 catalytic activity. Extracellular LOXL2 also interacts with HSP90, although the functional consequences of this interaction are unknown. Nuclear LOXL2 is negatively regulated by the lncRNA GATA6-AS. The question mark (?) means that the functional consequences of LOXL2 phosphorylation are unknown.

**Figure 4 ijms-24-14405-f004:**
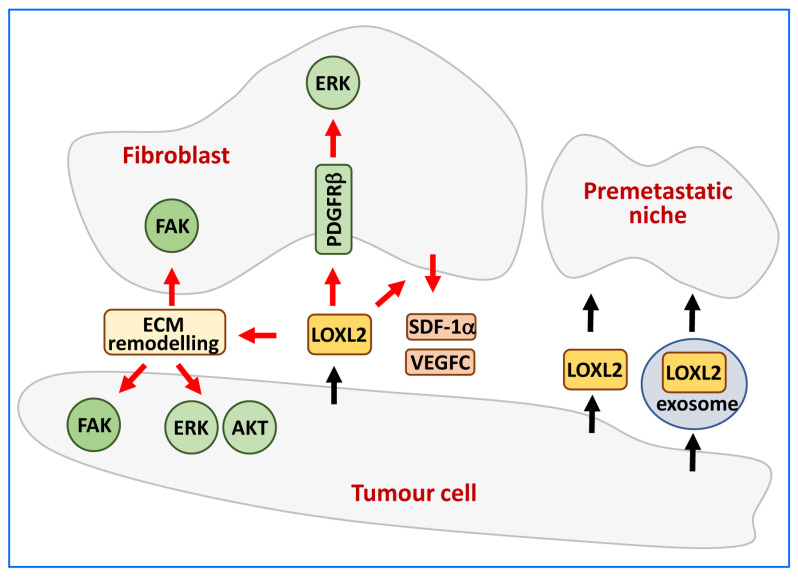
Targets of secreted LOXL2. LOXL2 provokes ECM remodelling, activating the FAK signalling pathway in fibroblasts and tumour cells. Additionally, it stimulates the AKT and ERK signalling pathways specifically in tumour cells. LOXL2 oxidises PDGFRβ, enhancing ERK signalling in fibroblasts, and increases the secretion of lymphangiogenic factors (VEGFC and SDF-1α). In distant organs, secreted or exosomal LOXL2 stimulates the formation of premetastatic niche.

**Figure 5 ijms-24-14405-f005:**
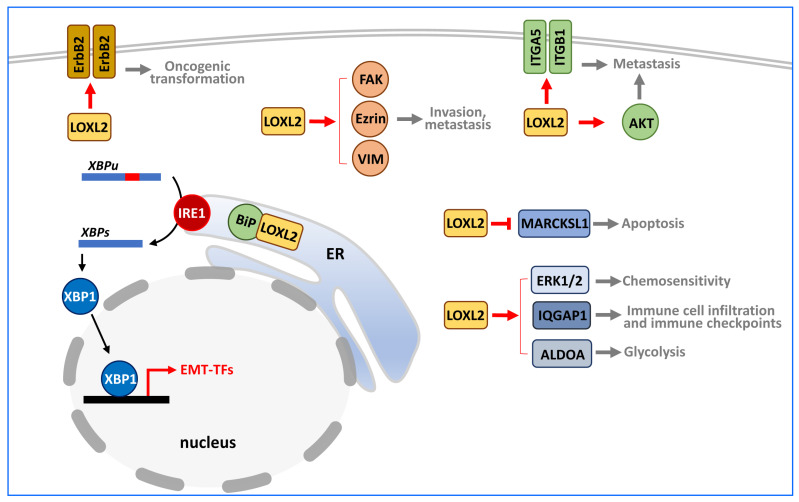
Intracellular targets of LOXL2. LOXL2 can influence numerous pro-tumorigenic actions in different tumour contexts by interacting with various effectors located in the plasma membrane (i.e., ERBB2 receptor and ITGA5/ITGB1 integrins) or the cytoplasm (FAK, ezrin, VIM, AKT, MARKSL1, ERK1/2, IQGAP1, and ALDOA), thereby affecting diverse cellular processes. The interaction of LOXL2 and HSPA5/BiP in the ER leads to activation of the transcription factor XBP1, which upregulates several EMT-TFs. Red arrows indicate positive regulation, and red blunt-end arrows signify negative regulation exerted by LOXL2 on the indicated targets. The final functional processes altered by LOXL2 action are marked in grey.

**Figure 6 ijms-24-14405-f006:**
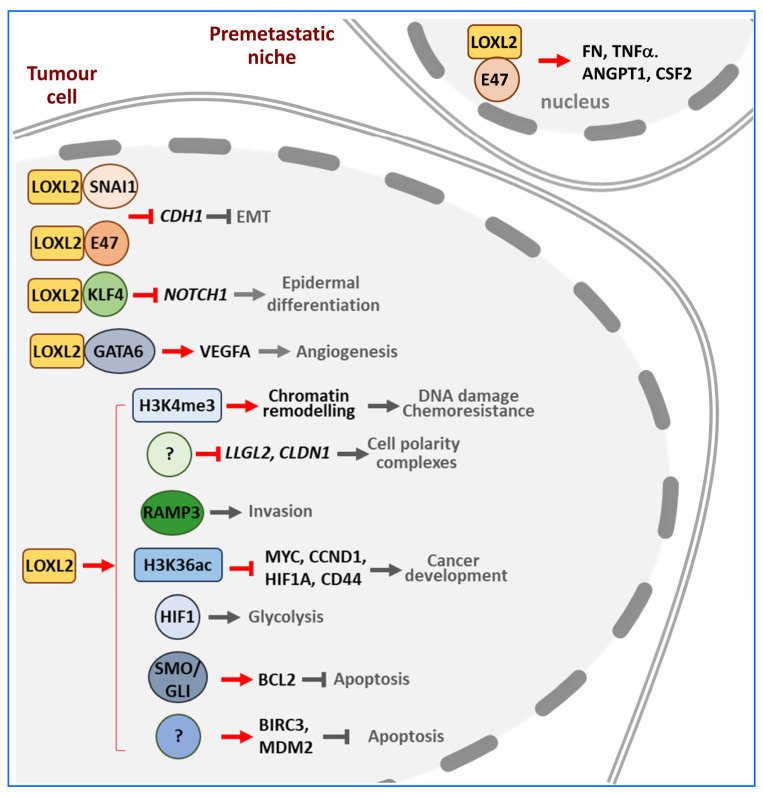
Nuclear targets of LOXL2. In different tumour scenarios, nuclear LOXL2 exerts its pro-tumorigenic roles by interacting with various transcription factors (SNAI1, E47, KLF4 and GATA6), modifying histone marks (H3K4me3 and H3K36ac), and upregulating the expression of different effectors (HIF1, SMO/GLI, and RAMP3). The downregulation of cell polarity complex genes (LLGL2, CLDN1) and upregulation of antiapoptotic genes (BIRC3 and MDM2) are mediated by unknown transcription factors. Red arrows denote positive regulation and red blunt-end arrows signify negative regulation exerted by LOXL2 on the indicated targets. Final functional processes altered by LOXL2 action are marked in grey. The question mark (?) means that the direct LOXL2 target or the functional consequences of an LOXL2 action are unknown.

**Figure 7 ijms-24-14405-f007:**
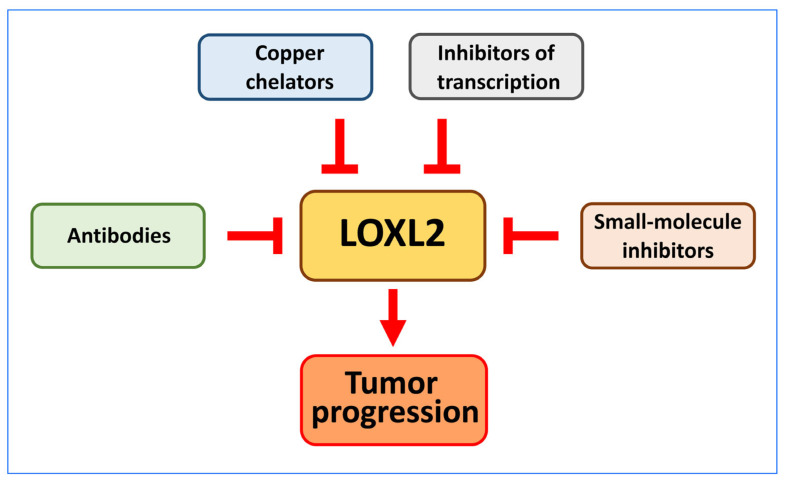
Strategies used to block LOXL2 action on tumour progression. The generation of anti-LOXL2 antibodies, optimisation of copper chelators, search for natural products capable of blocking the expression of LOXL2, or the development of small molecules designed to inhibit the catalytic activity of LOXL2 are different approaches currently being developed with the ultimate goal of interfering with the pro-tumorigenic action of LOXL2.

**Table 1 ijms-24-14405-t001:** Factors controlling *LOXL2* gene transcription.

Factors	Ref.
Cancer-associated fibroblasts	[61]
ETS transcription factor ELK3 (ELK3)	[62]
Transcription factor forkhead box A1 (FOXA1)	[63]
Glial cell-line derived neurotrophic factor (GDNF)	[64]
DNA replication GINS complex, subunit 2 (GINS2)	[65]
Oncostatin M	[39,66]
37/67 kDa laminin-1 receptor ribosomal protein SA (RPSA)	[67]
Histone methyltransferase, SET, and MYND domain containing 3 (SMYD3)	[68]
Succinate dehydrogenase complex iron sulphur subunit B (SDHB)	[69]
Transcription factor SP1	[70,71]
Transforming growth factor beta (TGFβ)	[72]
Vitamin D	[73]
Hepatitis transactivator protein X (HBx)	[74]
Hepatitis C virus core protein	[75]

**Table 2 ijms-24-14405-t002:** Genetically engineered mouse models of lysyl oxidases.

Gene	Genetic Model	Phenotype	Ref.
Lox	Lox KO	Alteration in cardiovascular and respiratory systems. Perinatal lethality	[129,130]
	Lox KI	Alteration in vascular remodelling	[131]
Loxl1	Loxl1 KO	Pelvic prolapse	[132,133]
		Sex-linked skeletal alterations	
Loxl2	Loxl2 KO	Congenital heart defects. Hepatic vessel distention. Incomplete perinatal lethality	[122]
	Loxl2 KO	Female uterine hyperplasia	[17]
	Conditional LOX2 KO (adult)	Stress induced cardiac fibrosis and increased cardiac injury	[133]
	Loxl2 KI	Male sterility	[122]
	L2ΔE13 KI (splice variant)	Loss of adipose tissue	[16,134]
Loxl3	Loxl3 KO	Defects in muscle–skeletal and lung system development. Incomplete perinatal lethality	[135,136,137,138]
	Conditional Loxl3 KO (adult)	Progressive loss of hearing via Loxl3 ablation in inner ear	[139]
Loxl4	Loxl4 KO	No phenotype	[140]

**Table 3 ijms-24-14405-t003:** Genetically engineered mouse models of Loxl2 and Loxl3 related to cancer.

Gene	Genetic Model	Cancer Model	Phenotype	Ref.
Loxl2 constitutive	Loxl2 KO	DMBA/TPA mouse skin carcinogenesis	Decreased tumour burden and malignant progression	[122]
	Loxl2 KI	DMBA/TPA mouse skin carcinogenesis	Decreased latency, increased tumour burden, and malignant progression	[122]
	Loxl2 KO	Spontaneous uterine cancer	Uterine hyperplasia and uterine carcinomas	[17]
	L2ΔE3 KI	Oesophageal cancer	Metabolic reprograming	[16]
Loxl2 conditional	Loxl2 KO (mammary glands)	MMTV-PyMT-breast cancer	Decreased lung metastasis	[142]
	Loxl2 KI (mammary glands)	MMTV-PyMT-breast cancer	Increased lung metastasis	[142]
	Loxl2 KO (pancreatic tumours)	KPC (Kras/Tp53/Pdx1-Cre)-L2-KO/ KC (Kras/Pdx1-Cre)-L2-KO	Decreased metastasis, increased overall survival. Alteration in collagen crosslinking	[39]
	Loxl2 KI (pancreatic tumours)	KPC (Kras/Tp53/Pdx1-Cre)-L2-KI/ KC (Kras/Pdx1-Cre)-L2-KI	Increased metastasis and tumour growth, decreased overall survival. Induction of EMT and stemness.	[39]
Loxl3 conditional/ inducible	Loxl3 (melanoma)	Tyr-Cre^ER^/Braf/Pten/L3-KO	Decreased tumour burden and reduced lymphatic dissemination	[143]

## Data Availability

Not applicable.

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
