# Peer review of "LOXL2 in Cancer: A Two-Decade Perspective"

_ijms, 2023, doi:10.3390/ijms241814405_

Round 1

Reviewer 1 Report

This is a very thorough review on the role of LOXL2 in cancer. It is well written and designed.

My major concern is about the actual role of LOXL2 in cancer. A similar review could be written for LOX and the authors already included data for LOXL3. the big question is whether a selective inhibition of LOXL2 given the redundancy in the system will suffice to improve cancer treatments. I do not expect the authors to answer the question but would propose a critical assessment at the end.

There are some specific issues that I would like to raise:

Line 41 Demethylation of trimethylated lysine 4 in histone H3 trough the catalytic (LTQ) domain is a very unusual mechanism. As it has been reported by a single group primarily, I would suggest to mention the challenges in the concept. 

Line 141: the association of AA alignment and impact on function is poor highly speculative. Either there should be evidence or clearly labelled a speculation based on low AA alignment.

Throughout the intracellular and intranuclear mechanisms it is not clear what actually LOXL2 is doing. eg line 327 the authors do not even state that LOXL2 phophorylates AKT but that genes leading to this phosphorylation are upregulated. It would be worthwhile to clearly separate direct and indirect effect and label them accordingly. 

Line 555 The new ELISA assay is supposed to be promising but the concentrations of the neo-epitope are 100x higher than LOXL2 concentrations measured in serum and cited by the authors. I would be cautious about any statements on the usefulness.

line 577: AB0023 is actually not inhibiting the enzymatic function: An activity‐based bioprobe differentiates a novel small molecule inhibitor from a LOXL2 antibody and provides renewed promise for anti‐fibrotic therapeutic strategies - PMC (nih.gov) and this finding should be discussed 

line 619: The most selective  inhibitors for LOXL2 are PAT-1251 and PXS-2A, while PXS-5153 and PXS-3A are both LOXL2/3 inhibitors. All of these should be mentioned.

Author Response

Reviewer 1.

We thank the reviewer for their insightful comments that have greatly contributed to an increase in the quality of our manuscript.

All the specific comments raised by the reviewer have been addressed.

My major concern is about the actual role of LOXL2 in cancer. A similar review could be written for LOX and the authors already included data for LOXL3. the big question is whether a selective inhibition of LOXL2 given the redundancy in the system will suffice to improve cancer treatments. I do not expect the authors to answer the question but would propose a critical assessment at the end.

We thank the reviewer for the positive remarks on the ms. We also appreciate the comment on the actual role of LOXL2 in cancer and the potential redundancy with other lysyl oxidases. Nevertheless, we consider that the present evidence supports a relevant role of LOXL2 in several types of cancer as summarized in the present review. Attending to the reviewer’s comment we have included a critical assessment on this relevant issue at the end of the ms (line 742 of the revised version). We have added the following sentence a “In addition, a better knowledge of the potential redundancy among other members of the lysyl oxidase family in specific types of cancer or defined tumour stages will improve the therapeutic options associated with LOXL2 in the near future”.

There are some specific issues that I would like to raise:

Line 41 Demethylation of trimethylated lysine 4 in histone H3 trough the catalytic (LTQ) domain is a very unusual mechanism. As it has been reported by a single group primarily, I would suggest to mention the challenges in the concept.

  1. We have added the following sentence “However, it should be noted that this deamination activity poses a conceptual challenge that needs further confirmation.” (line 43 of the revised version).

Line 141: the association of AA alignment and impact on function is poor highly speculative. Either there should be evidence or clearly labelled a speculation based on low AA alignment.

  1. As suggested by the reviewer we have modified the sentence. The new sentence is as follows: “This analysis revealed that 36.5 % (50 out 137) of the detected changes occur in fully conserved residues, leading us to speculate that they could potentially impact LOXL2 activity (Figure S1A-G). However, experimental evidence will be necessary to validate this hypothesis due to the low amino acid alignment between the LOXL2, LOXL3 and LOXL4 proteins”. (lines 146-151 of the revised version).

Throughout the intracellular and intranuclear mechanisms it is not clear what actually LOXL2 is doing. eg line 327 the authors do not even state that LOXL2 phophorylates AKT but that genes leading to this phosphorylation are upregulated. It would be worthwhile to clearly separate direct and indirect effect and label them accordingly.

We agree with the reviewer’s observation. In certain sentences it is not clear whether the effect of LOXL2 is direct or indirect. This ambiguity has been addressed and clarified in the revised version. Consequently, we have modified the following sentences in the revised manuscript:

Line 329:”In normal mammary epithelial cells, LOXL2 overexpression induces oncogenic transformation and cancer progression by activating the Erb-B2 receptor tyrosine kinase 2 (ERBB2) through the production of reactive oxygen species (ROS) [107].”

Line 335: “Additionally, LOXL2 can promote tumour metastasis by regulating the levels of phosphorylated AKT through mechanisms that are not yet fully understood [110] and by stabilizing integrin subunits α5 (ITGA5) and β1 (ITGB1) [111]”.

Line 339: “LOXL2 also affects endoplasmic reticulum (ER) homeostasis. Its overexpression indirectly activates the unfolded protein response (UPR) leading to epithelial-to-mesenchymal transition (EMT).

Line 350: “LOXL2 can also modulate other cancer hallmarks. It influences tumour chemosensitivity by enhancing autophagy. LOXL2 increases ATG7 expression possibly by promoting the phosphorylation of ERK1/2 through an unknown mechanism [117].

Line 555 The new ELISA assay is supposed to be promising but the concentrations of the neo-epitope are 100x higher than LOXL2 concentrations measured in serum and cited by the authors. I would be cautious about any statements on the usefulness.

As suggested by the reviewer we have modified the sentence. The new sentence is as follow:

“This novel and specific ELISA assay appears to be another approach for detecting elevated LOXL2 levels in fibrosis or cancer. However, it is worth noting that the high concentration of the neo-epitope required for detection in the ELISA assay raises a cautionary note about its current utility” (line 571 of the revised version).

line 577: AB0023 is actually not inhibiting the enzymatic function: An activity‐based bioprobe differentiates a novel small molecule inhibitor from a LOXL2 antibody and provides renewed promise for anti‐fibrotic therapeutic strategies - PMC (nih.gov) and this finding should be discussed.

We thank the reviewer for bringing to our attention this important point regarding the comparison of the novel small molecule inhibitor with the anti-LOXL2 AB0023 antibody and, accordingly, we have included in the revised version the relevant study by Findlay et al. 2021 (reference 166 in the revised version). However, we respectfully disagree with the reviewer’s comment “AB0023 is actually not inhibiting the enzymatic function”. Rodriguez et al. (JBC 2010) characterized AB0023 as an inhibitor of LOXL2 enzymatic activity in vitro by a non-competitive mechanism and described AB0023 binding to the 4th SRCR, supporting it as an allosteric inhibitor. To our knowledge, this biochemical characterization of AB0023 has not been questioned in the field until present. Therefore, we think relevant to maintain that information in the present review. Unfortunately, the reference by Rodriguez et al, was not properly cited when describing the AB0023 antibody in the original version of the ms. (Section 6.1. Antibodies), but in another part of the same section, a fact that could be misleading. We deeply regret this mistake and we have now included the proper citation it the revised version of the manuscript (reference 156, line 596 of the revised version).

The revised version of this section also includes additional clarification on the lack of complete inhibition of LOXL2 enzymatic activity by AB0023 and correct citations of AB0023 and humanized Simtuzumab antibody when corresponding. Apart from inclusion of the novel study of Findlay et al, comparing the novel small molecule inhibitor with the anti-LOXL2 AB0023 antibody, mentioned above.

Please, see complete revised part of this section, lines 599-616 of the revised ms

line 619: The most selective inhibitors for LOXL2 are PAT-1251 and PXS-2A, while PXS-5153 and PXS-3A are both LOXL2/3 inhibitors. All of these should be mentioned.

We also thank the reviewer for this relevant comment. In trying to refer mainly to cancer in this review we avoided references to other diseases such as fibrosis. We have now introduced the description of the suggested inhibitors (PAT-1251 and PXS-5153) not covered in the original version of the ms. (see lines 665-670 of the revised version). Regarding PXS-3A inhibitor, after a thorough review of the bibliography we have been unable to find any information on that compound (either as PXS-3A or PXS-S3A) in published literature from public databases (Pubmed, etc…). We regret not being able to include data from the PXS-3A compound in the revised version.

Reviewer 2 Report

This is an extensive and thoroughly covered review paper in many aspects.It is impressive that the authors seem to gather the most current literature concerning LOXL2 and summarized the essence of these studies. This is invaluable for those who are in the field, as well as for the general audience.

I have a few comments that need to be addressed before recommending this manuscript to be published.

Scientific comments

1.     Line 35-36: “LTQ undergoes a spontaneous auto-catalysed posttranslational reaction that covalently links a lysine and a tyrosine residue (K653 and Y689 in LOXL2).” This sentence needs to be revised.It has not been defined whether the biogenesis of LTQ is spontaneous nor auto-catalyzed for the LOX-family of proteins. It was suggested for the fruit fly lysyl oxidase-like protein. The crosslink of K653 occurs after Y689 being oxidized to a quinone.

2.     The first 3D-predicted structure of the active site of LOXL2 based on the precursor structure and solution study has been published in this journal that should be included (mentioned) after line 64 in this review. 

3.     Line 237-239: This sentence is misleading. The processing of the first two SRCR domains is not essential for the amine oxidase activity in solution or crosslinking of collagen type IV in ECM secreted from fibroblasts in reference 90.

4.     It will be better to include a Table for inhibitors (covered in sections 6.3 and 6.4) that have been synthesized or identified with their names, structures and IC50 with the corresponding citation.

5.     Line 646-648: “Common to many of the small molecules developed as LOXL2 inhibitors is the presence of the primary amine that will compete with lysine in the interaction and reaction with the LTQ cofactor in the active site of LOXL2, allowing for specific binding to the enzyme.” This sentence needs to be revised.

Common to  A common feature of the

the primary amine  the primary amino group

lysine  epsilon amino group of Lys residues in LOXL2 substrates

interaction and reaction  omit one of them (I do not know why they need to differentiate interaction and reaction)

Minor issues

1.     Throughout text: lysine tyrosyl quinone  lysine tyrosylquinone

2.     Line 32: omit “histidine-rich”  

3.     Line 37: oxidation  oxidative deamination

4.     Line 43: specify “different” proteases

5.     Throughout the text: “cues” is not an appropriate scientific term and it should be replaced by factors or signaling pathways.

6.     Throughout the text: the authors should be consistent with one letter amino acid abbreviation.

7.     Line 78: replace “eliminating” with “a loss of”

8.     Line 136: cooper  copper, replace “formation” with “precursor”

9.     Line 183: omit “includes mechanisms that” control  controls

10.  Line 653: omit “a specific lysine residue”

11.  Under 7 Future perspectives

Line 694: The initial description  the first description or first mention

Line 695: many cancer types  many types of cancers

Line 696: Currently, we are aware of  Currently, it is known…

Line 697: its effects  what does “its” refer to?

Line 698: cancer fitness  fitness should be replaced with a more appropriate word.

            Line 698: we know that  omit

            Line 703: each of the different catalytic domains of the enzyme – catalytic domain refers                                    to the C-terminal amine oxidase domain. The authors may intend to refer     to SRCR domains in addition to the C-terminal amine oxidase domain, if so, they should   just spell out each of the SRCR domains and C-terminal amine oxidase domain.

            Line 704: attenuate enhanced LOXL2 expression  control the LOXL2 expression level

Author Response

Reviewer 2.

We thank the reviewer for their insightful comments that have greatly contributed to an increase in the quality of our manuscript.

All the specific comments raised by the reviewer have been addressed.

Scientific comments

  1. Line 35-36: “LTQ undergoes a spontaneous auto-catalysed posttranslational reaction that covalently links a lysine and a tyrosine residue (K653 and Y689 in LOXL2).” This sentence needs to be revised.It has not been defined whether the biogenesis of LTQ is spontaneous nor auto-catalyzed for the LOX-family of proteins. It was suggested for the fruit fly lysyl oxidase-like protein. The crosslink of K653 occurs after Y689 being oxidized to a quinone.
  2. The sentence has been changed to “LTQ cofactor is formed from conserved lysine and tyrosine residues by posttranslational modification (K653 and Y689 in LOXL2). (lines 35-36 of the revised version).

  1. The first 3D-predicted structure of the active site of LOXL2 based on the precursor structure and solution study has been published in this journal that should be included (mentioned) after line 64 in this review.

We deeply regret this omission and we have now included the proper citation it the revised version of the ms. The new sentence (lines 65-66 of the revised version) is as follow: “A low-resolution structure of the full-length LOXL2 obtained by X-ray scattering and electron microscopy [21], the crystal structure of a precursor form at a resolution of 2.4 Å [22] and a 3D-predicted structure of the C-terminal amine oxidase domain of LOXL2 [23] are currently available. (reference 23: Meier, Kuczera & Mure. Int. J. Mol. Sci. 2022, 23(21), 13385; https://doi.org/10.3390/ijms232113385).

  1. Line 237-239: This sentence is misleading. The processing of the first two SRCR domains is not essential for the amine oxidase activity in solution or crosslinking of collagen type IV in ECM secreted from fibroblasts in reference 90.

We thank the reviewer for bringing this important and controversial point to our attention regarding the role of PACE4 proteolytic processing in the amine oxidase activity of LOXL2. Indeed, there is conflicting evidence in the literature on this matter. One study suggests that proteolytic processing is not crucial for the amine oxidase activity in solution or for the crosslinking of collagen type IV in the extracellular matrix (ref. 90), while another study contends that it does not impact LOXL2-mediated crosslinking of soluble collagen type IV in vitro but is essential for crosslinking insoluble collagen IV within the ECM (ref. 91). We deeply regret the initial oversight and we have now included this controversial point it the revised version of the ms. The new sentence (lines 245-250 of the revised version is as follow: “The importance of this proteolytic processing on the amine oxidase activity of LOXL2 is controversial. One report suggests that the proteolytic processing is not essential for the amine oxidase activity in solution or crosslinking of collagen type IV in ECM [90] while another report suggests that it does not affect LOXL2-mediated crosslinking of soluble collagen type IV in vitro but is essential to crosslink insoluble collagen IV within the ECM [91].

  1. It will be better to include a Table for inhibitors (covered in sections 6.3 and 6.4) that have been synthesized or identified with their names, structures and IC50 with the corresponding citation.
  2. Table has been included (Table S1 of the revised version) and have been introduced in lines 644-645.

  1. Line 646-648: “Common to many of the small molecules developed as LOXL2 inhibitors is the presence of the primary amine that will compete with lysine in the interaction and reaction with the LTQ cofactor in the active site of LOXL2, allowing for specific binding to the enzyme.” This sentence needs to be revised.

Common to  A common feature of the

the primary amine  the primary amino group

lysine  epsilon amino group of Lys residues in LOXL2 substrates

interaction and reaction  omit one of them (I do not know why they need to differentiate interaction and reaction)

  1. The sentence has been changed to: “A common feature of the small molecules developed as LOXL2 inhibitors is the presence of a primary amino group. This group competes with the epsilon amino group of lysine residues in LOXL2 substrates during their interaction with the LTQ cofactor, enabling specific binding to the enzyme. (Lines 677-681 of the revised version)

Minor issues

  1. Throughout text: lysine tyrosyl quinone  lysine tyrosylquinone
  2. Corrected (lines 11 and 33 of the revised version)

  1. Line 32: omit “histidine-rich”
  2. Deleted (line 32 of the revised version)

  1. Line 37: oxidation  oxidative deamination
  2. Corrected. Line 39 of the revised version
  3. Line 43: specify “different” proteases
  4. Proteases have been specified. Line 46 of the revised version.

  1. Throughout the text: “cues” is not an appropriate scientific term and it should be replaced by factors or signaling pathways.
  2. Replaced. Line 73 and 163 of the revised version.

  1. Throughout the text: the authors should be consistent with one letter amino acid abbreviation.
  2. revised

  1. Line 78: replace “eliminating” with “a loss of”
  2. Replaced. Line 83 of the revised version.

  1. Line 136: cooper  copper, replace “formation” with “precursor”
  2. Corrected and Replaced. Line 142 of the revised version.

  1. Line 183: omit “includes mechanisms that” control  controls
  2. Omitted. Line 191 of the revised version.

  1. Line 653: omit “a specific lysine residue”
  2. Omitted. Line 685 of the revised version.

  1. Under 7 Future perspectives

Line 694: The initial description  the first description or first mention

  1. Replaced. Line 727 of the revised version

Line 695: many cancer types  many types of cancers

  1. Replaced. Line 728 of the revised version.

Line 696: Currently, we are aware of  Currently, it is known…

  1. Replaced. Line 729 of the revised version.

Line 697: its effects  what does “its” refer to?

  1. “its” have been changed to “LOXL2 effects”. Line 731 of the revised version.

Line 698: cancer fitness  fitness should be replaced with a more appropriate word.

  1. Changed to “capabilities”. Line 732 of the revised version.

            Line 698: we know that  omit

  1. Omitted. Line 732 of the revised version.

            Line 703: each of the different catalytic domains of the enzyme – catalytic domain refers                                    to the C-terminal amine oxidase domain. The authors may intend to refer     to SRCR domains in addition to the C-terminal amine oxidase domain, if so, they should   just spell out each of the SRCR domains and C-terminal amine oxidase domain.

  1. The new sentence is as follow: “It may be necessary to utilize distinct therapeutic drugs designed to inhibit the SRCR domains in addition to the C-terminal oxidase domains of the enzyme” (Lines 736-738 of the revised version)

            Line 704: attenuate enhanced LOXL2 expression  control the LOXL2 expression level

  1. Changed. (lines 738 of the revised version).

Round 2

Reviewer 1 Report

I disagree with the impact of LOXL2 inhibitors for the treatment of cancer and the review provides a very positive view on the therapeutic opportunity. However, this should not stop its publication.

Author Response

Reviewer #1

I disagree with the impact of LOXL2 inhibitors for the treatment of cancer and the review provides a very positive view on the therapeutic opportunity. However, this should not stop its publication.

We thank to the reviewer for his/her comment about the impact of LOXL2 inhibitors for the treatment of cancer. As we described in the 'future perspectives' section, we believe that there are numerous challenges and difficulties to overcome in the development of effective therapeutic agents focused on LOXL2, being the possible functional redundancy between different members of the family one of them. Perhaps the last sentence of the manuscript does not clearly reflect this view; consequently, we have modified it. Now the sentence is as follows:

“In addition, the potential functional redundancy among different LOX proteins may require the inhibition of additional family members alongside LOXL2 to enhance the therapeutics strategies related to this family of proteins”.